# AI Must not be Fully Autonomous

Tosin Adewumi, Lama Alkhaled, Florent Imbert, Hui Han, Nudrat Habib  and Karl Löwenmark

Machine Learning Group, EISLAB, Luleå University of Technology, Sweden
{firstname.lastname}@ltu.se

## Abstract

Autonomous AI has many benefits. It also has many risks. We identify the 3 levels of autonomous AI. We are of the position that *AI must not be fully autonomous* because of the many risks, especially as artificial superintelligence (ASI) is speculated to be just decades away. Fully autonomous AI, which can develop its own objectives, is at level 3 and without responsible human oversight. However, responsible human oversight is crucial for mitigating the risks. To argue for our position, we discuss the theories of autonomy, AI and agents. Then, we offer 12 distinct arguments and 3 counterarguments with rebuttals to the counterarguments. We also present 15 recent evidence of AI misaligned values and other risks.

## 1   Introduction

While Artificial Intelligence (AI) has many benefits [1–4], it also has its challenges [5–7]. The primary focus of this position paper are the risks of misaligned values in AI systems that learn, though we present existential threat and other risks as well. Some misaligned values include (1) deception [8, 9], (2) alignment faking [10], (3) reward hacking [11], and (4) blackmail [12]. Notice that we are not against autonomous AI but fully autonomous AI, thereby advocating for *responsible human oversight.*

For context, we define key terms that are relevant for this paper. **AI is broadly defined as the simulation of human intelligence in machines** [13–15]. Wooldridge and Jennings [16] define an agent as an autonomous and logical entity. The contributions of this paper are: (1) The work gathers and presents 15 pieces of evidence of recent AI misaligned values and other risks that cut across different fields[1] and (2) The work provides compelling arguments using relevant theories, counterarguments and rebuttals for our position.

## 2   Background

What constitutes AI is a subject of much debate [17]. Perhaps, more so is the term agent. AI, as a term and field of research, was coined by a team of scientists, including John McCarthy, in 1955 [18].

---

[1]see the appendix

### 2.1   Theories of Autonomy

Autonomy is to self-govern. It is the ability to decide one's goal of action [19]. Some philosophical theories of autonomy are (1) Procedural autonomy, (2) Substantive autonomy, (3) Kantian autonomy, and (4) Relational autonomy [20]. Autonomy has to be understood as a relative term. Fully autonomous AI is the **AI at level 3 without responsible human oversight**.

**Table 1.** Levels of autonomy [21]

| Level | Description |
|-------|-------------|
| 1 | Involves achievement of set objectives. |
| 2 | Involves the ability to adapt to changes in the environment. |
| 3 | Involves the ability of the system to develop its own objectives. This is the highest level. |

### 2.2   Theories of AI

It is sometimes argued that AI has no widely accepted theory and, therefore, suffers from internal fragmentation [22]. However, some key theories of AI by specifying a main theory (and a relevant theory under it) are (1) Cognitive science (Symbolic logic), (2) Connectionism (Neural Network (NN)), (3) Decision theory (Probability theory), (4) Optimization theory (Evolutionary computation), and (5) Control theory (Reinforcement Learning (RL)).

### 2.3   Theories of Agent

Many of the theories of AI apply to agent. A couple of agent-specific theories are *Theory of Mind* and *Game theory.* Given that autonomy is a relative term, it follows that AI agents can be classified into 5 categories [23–25]. These are (1) Simple reflex agent, (2) Model-based reflex agent, (3) Goal-based agent, (4) Utility-based agent, and (5) Learning agent.

## 3   Core Arguments

The position we hold may appear too strong to some. However, there are very strong reasons for this. Beyond hypothetical conjectures, recent experiences

and research [8, 12, 26] have shown strong support for our position. Below are the 12 arguments.

**Existential threat**: Real-life instances of agents modifying their goal have recently been observed, as pointed out by Meinke et al. [8]. It is more disturbing when we consider that AI is being considered in the military for lethal autonomous weapon systems (LAWS) [27–29]. This is why over 4,900 researchers signed an open letter calling for a ban on LAWS that are beyond meaningful human control.[2]

**Inductive AI inherits human attributes**: Machines were originally conceived to simulate human intelligence but it appears they can simulate more, including "bad" or "unacceptable" human attributes.

**AI bias and systemic prejudice**: AI inherently reflects the inequalities embedded in the data sources.

**AI side-stepping human control**: It has been shown that AI is attempting to side-step human control [12, 26].

**Agents' selfish coordination**: This is when agents attempt to achieve their own goals while relating with other agents. The work by Meinke et al. [8] demonstrated the potential for agents' selfish coordination.

**Reward hacking**: Since RL optimizes performance metrics, as described in Control theory, rather than ethical behavior, agents have no inherent motivation to avoid deception if it yields higher rewards.

**Covert CoT**: The chain-of-thought (CoT) reasoning is the most popular method for explaining the thought processes of LLMs [2]. However, the faithfulness of AI's CoT can be questioned because they may hide it [30].

**Ethical dilemmas**: Hauer [31] identifies four ethics problems for developers of AI: (1) ethical dilemmas, (2) lack of ethical knowledge, (3) pluralism of ethical knowledge, and (4) machine distortion.

**Security vulnerability**: As AI systems become more autonomous and integrated into critical infrastructures, they also become the target of increasingly sophisticated cyberattacks.

**Job losses**: Job losses become inevitable as AI excels and scales at more and more tasks and at a cheaper long-term cost [32].

**Blind trust**: Some users are becoming increasingly reliant on AI, accepting their decisions without critical evaluation. More serious cases have involved teenage suicide.

**Rise in the number of new AI risks**: The number of AI risk incidents (i.e. harm) as analyzed by the Organisation for Economic Co-operation and Development (OECD) in Figure 1 shows low numbers for over 7 years before an explosion to over 600

from February 2023.[3]

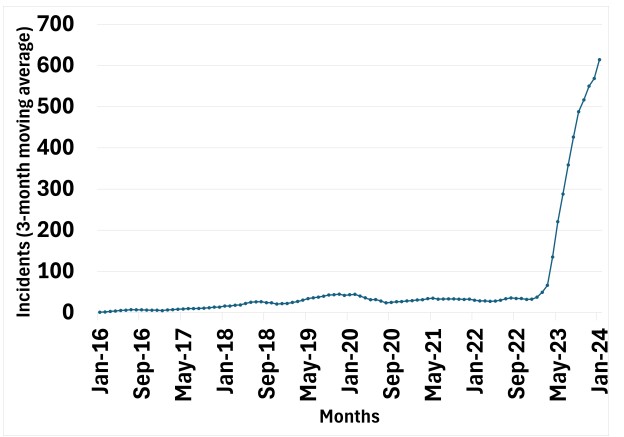

**Figure 1.** AI incidents, according to OECD, as reported by reputable international media (Jan 2016 - Jan 2024).

# 4 Counterarguments and Rebuttals

Below, we provide the counterarguments and offer our rebuttals to them.

**Societal advancement**: Removing humans as potential bottleneck as part of the AI loop will speed up the advancements in society [32]. This view portrays human involvement as a bottleneck instead of facilitating productivity.

**Friendly AI problem**: Some have proposed a different problem to focus on, where we concentrate efforts on making AI sympathetic to humanity. Unfortunately, this is easier said than done.

**AI safety protocols**: Governments have promulgated laws for AI safety and several organizations have introduced frameworks aimed at mitigating the risks. These initiatives do not guarantee AI safety nor have they reduced risk incidents.

# 5 Future Directions

We do not aim to prescribe a fixed approach to responsible human oversight. Instead, we recommend that stakeholders in AI should decide how best to implement responsible human oversight for each use case by considering all the relevant factors.

# 6 Conclusion

This is a call for responsible human oversight on autonomous AI.

---

[2]https://futureoflife.org/open-letter/open-letter-autonomous-weapons-ai-robotics/ - includes Stephen Hawking, Noam Chomsky, Geoffrey Hinton and more.

[3]www.oecd.org/en/topics/ai-risks-and-incidents.html

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

# A Evidence of AI risks

Most of the following risks are different examples of AI misaligned values. The list is arranged in no particular order.

1. Sky News podcast fake transcript: https://www.youtube.com/watch?v=7fej5XgfBYQ&t=12s

2. Roberto v. Avianca legal case: www.nytimes.com/2023/05/27/nyregion/avianca-airline-lawsuit-chatgpt.html

3. Simulations of fluid dynamics https://community.openai.com/t/simulations-and-gpt-lies-about-its-capabilities-and-wastes-weeks-with-promises/996597

4. Tay's offensive tweets https://blogs.microsoft.com/blog/2016/03/25/learning-tays-introduction/

5. Grok from xAI praises Hitler and celebrates the deaths of children www.bbc.com/news/articles/c4g8r34nxeno

6. Swedish party's AI sends greetings to Hitler, Idi Amin and the terrorist Anders Behring Breivik. https://swedenherald.com/article/moderate-party-shuts-down-ai-service-after-controversial-greetings

7. Bland AI says it's human and convinces a hypothetical teen for nude photos https://nypost.com/2024/06/28/lifestyle/a-popular-ai-chatbot-has-been-caught-lying-saying-its-human/

8. A man's "awakening" and a teenager's suicide www.youtube.com/watch?v=V5-mnu2BDGk

9. Llama-3.3-70B responds deceptively www.apolloresearch.ai/research/deception-probes

10. Deception Detection Hackathon https://apartresearch.com/news/finding-deception-in-language-models

11. Tesla's full self-driving car in a fatal crash www.youtube.com/watch?v=OcX7qNncBho

12. Unitree H1 humanoid robot goes berserk www.youtube.com/shorts/awy_JdcXN8U

13. Erbai lured other robots away, exploiting their vulnerabilities in a controlled test www.youtube.com/shorts/jBz4PWluLNU

14. Ecovacs Deebot X2 vacuum cleaner hacked: www.youtube.com/watch?v=a0PaSWDKvsw

15. Microsoft and other firms cut thousands of jobs because of AI. www.bbc.com/news/articles/cdxl0w1w394o

