# OpenReview forum: "AI Must not be Fully Autonomous"
_NLDL.org/2026/Abstracts_Track — NLDL 2026 Abstracts_

### Official Review · Reviewer_h4wf · 2025-10-26

**Soundness:** 3
**Correctness:** 4
**Rating:** 5
**Confidence:** 5

**Summary:**

This position paper argues that AI must not be fully autonomous, defined as Level 3 AI capable of developing its own objectives without responsible human oversight. The authors identify three levels of autonomous AI and, to support their position, offer 12 distinct arguments and 15 recent pieces of evidence of AI misaligned values and other risks, such as deception, reward hacking, and side-stepping human control. The core recommendation is a call for responsible human oversight on autonomous AI, with stakeholders deciding on the best implementation for each use case.

**Strengths:**

The abstract presents a clear, relevant, and highly contemporary position on a critical topic for the Machine Learning (ML) community: the risks of fully autonomous AI. This subject is highly likely to spark interesting and vital discussions at the conference.

    Strong, Evidence-Based Argument: The paper doesn't rely solely on conjecture but incorporates a dedicated section listing 15 recent pieces of evidence of AI risks, cutting across different fields (e.g., legal cases, deceptive models, fatal crashes). The sharp increase in AI incidents reported by the OECD, shown in Figure 1, provides a compelling visual argument for the urgency of the problem.

Comprehensive Risk Identification: The abstract thoroughly outlines various risks beyond existential threats, including misaligned values (deception, alignment faking, reward hacking, blackmail), ethical dilemmas, security vulnerability, and societal risks like job losses and blind trust.

Structured Discussion: The authors clearly structure their argument by defining key terms (AI, agent, autonomy), reviewing relevant theories, presenting core arguments, and offering three counterarguments with rebuttals. This logical flow makes the position easy to follow.

Timely Focus on "Fully Autonomous AI": The paper makes an important distinction between general autonomous AI and "fully autonomous AI" at Level 3 (developing its own objectives). This focus on the highest level of autonomy highlights the specific danger they aim to mitigate with human oversight.

**Weaknesses:**

While the core idea is sound and the evidence strong, a few elements could be clearer or more detailed to fully support the authors' contribution:

    Conciseness of Arguments: The abstract mentions 12 distinct arguments in the "Core Arguments" section. However, due to the abstract's page limit, the arguments on page 2 are presented in a highly abbreviated manner. For example, "Inductive AI inherits human attributes" and "AI bias and systemic prejudice" are only briefly stated. While understandable for an abstract, a few more sentences of elaboration on one or two of the key arguments would strengthen the paper's overall quality and detail.

Lack of Prescribed Oversight: In the "Future Directions" section, the authors state they do not aim to prescribe a fixed approach to responsible human oversight, recommending that stakeholders decide the best implementation per use case. While a flexible approach is practical, the abstract could benefit from providing a brief, high-level example or framework of what "responsible human oversight" might entail in a Level 3 AI context to make the main recommendation more concrete.

Citation for Levels of Autonomy: The three levels of autonomy are presented in Table 1. While a citation is provided for the table , the definition of the highest level—"Involves the ability of the system to develop its own objectives"—is a central element of the argument and could benefit from further context or philosophical grounding given the paper's discussion of autonomy theories.

---

### Official Review · Reviewer_mwVf · 2025-10-28

**Soundness:** 1
**Correctness:** 2
**Rating:** 2
**Confidence:** 4

**Summary:**

The authors propose a position paper arguing for restraining autonomy of AI systems, so that they never become fully autonomous.
In order to do this, they briefly touch upon 12 arguments in favour of their opinion and 3 counter-arguments.

**Strengths:**

This is a particularly timely topic, considering the fast progress made in AI in recent years with the introduction of LLMs.
It is also extremely relevant, since AI is no longer only accessible to experts, but is now available to everyone.

**Weaknesses:**

The abstract is poorly written. While there are no significant grammatical errors, the writing is superficial and not at the level required for a scientific publication, even a non-indexed one.
While the topic is relevant, and I agree with the position expressed by the authors, the way the argument is presented is weak.
I understand that this is simply an extended abstract and thus the space constraint does not allow for long and extended writing. However, the writing could have been structured better: instead of trying to cover all the arguments and counter-arguments, the authors could have focused on one or two of these points and provided a more in depth analysis.
This is particularly true for Section 4, in which the authors claim to provide a rebuttal for the counter-arguments to their thesis, but this is not really achieved. An argument cannot simply be dismissed by stating "this is easier said than done". This section also lacks references: for example, the authors argue that AI safety protocols "do not guarantee AI safety nor have they reduced risk incidents" wihtout providing any reference to back up this statement.
I do think that this is a relevant topic, but I believe that it is not properly argued in this submission. I would recommend the authors to either rewrite the extended abstract focusing on one or two arguments, or to extend this as a full-paper to properly cover all the presented arguments with adequate references.

---

### Official Review · Reviewer_3bc4 · 2025-11-03

**Soundness:** 2
**Correctness:** 2
**Rating:** 2
**Confidence:** 3

**Summary:**

The abstract identifies three levels of autonomous AI agents and the importance of identifying risks, as we are within decades of the development of artificial superintelligence. Building on that, the authors argue that human oversight is critical to mitigating the risks posed by autonomous AI agents. For which they provide 12 distinct arguments, 3 counterarguments, and 15 pieces of evidence of AI agents having misaligned values and other risks.

**Strengths:**

Strengths:
- The abstract addresses a highly relevant topic.
- The abstract also clearly sets up the premise (that AI must not be fully autonomous), its importance, and how they wish to defend their stance.

**Weaknesses:**

The abstract presents the work primarily as a position or discussion paper rather than a research study. While the topic is important and timely, the submission does not clearly articulate specific research questions or a defined analytical framework. As a result, it is not clear how the paper would make a concrete scientific contribution or advance our current understanding of the topic beyond a conceptual discussion.

---

### Decision · Program_Chairs · 2025-11-05

**Decision:**

Accept

**Comment:**

The reviewers found the abstract borderline, yet the PCs believe it will be of interest to the community and should have the opportunity be presented.